# The Physicochemical and Antibacterial Properties of Chitosan-Based Materials Modified with Phenolic Acids Irradiated by UVC Light

**DOI:** 10.3390/ijms22126472

**Published:** 2021-06-16

**Authors:** Beata Kaczmarek-Szczepańska, Marcin Wekwejt, Olha Mazur, Lidia Zasada, Anna Pałubicka, Ewa Olewnik-Kruszkowska

**Affiliations:** 1Department of Biomaterials and Cosmetics Chemistry, Faculty of Chemistry, Nicolaus Copernicus University in Torun, 87-100 Toruń, Poland; 289185@stud.umk.pl (O.M.); 296559@stud.umk.pl (L.Z.); 2Department of Biomaterials Technology, Faculty of Mechanical Engineering and Ship Technology, Gdańsk University of Technology, 80-008 Gdańsk, Poland; marcin.wekwejt@pg.edu.pl; 3Department of Laboratory Diagnostics and Microbiology with Blood Bank, Specialist Hospital in Kościerzyna, 83-400 Kościerzyna, Poland; apalubicka@op.pl; 4Physical Chemistry and Physicochemistry of Polymers, Faculty of Chemistry, Nicolaus Copernicus University in Torun, 87-100 Toruń, Poland; olewnik@umk.pl

**Keywords:** chitosan, phenolic acids, thin films, UVC light

## Abstract

This paper concerns the physicochemical properties of chitosan/phenolic acid thin films irradiated by ultraviolet radiation with wavelengths between 200 and 290 nm (UVC) light. We investigated the preparation and characterization of thin films based on chitosan (CTS) with tannic (TA), caffeic (CA) and ferulic acid (FA) addition as potential food-packaging materials. Such materials were then exposed to the UVC light (254 nm) for 1 and 2 h to perform the sterilization process. Different properties of thin films before and after irradiation were determined by various methods such as Fourier transform infrared spectroscopy (FTIR), scanning electron microscopy (SEM), atomic force microscopy (AFM), differential scanning calorimeter (DSC), mechanical properties and by the surface free energy determination. Moreover, the antimicrobial activity of the films and their potential to reduce the risk of contamination was assessed. The results showed that the phenolic acid improving properties of chitosan-based films, short UVC radiation may be used as sterilization method for those films, and also that the addition of ferulic acid obtains effective antimicrobial activity, which have great benefit for food packing applications.

## 1. Introduction

The detrimental activity of microbes of various types is one of the main reasons for the emergence of most human diseases. Bacterial, viral and fungal infections often result from lack or improperly performed decontamination processes, which remains in contradiction to the existing standards and regulations. Inadequate tools, air and various surfaces used when dealing with sterile tissues are the sources of pathogens [1]. Therefore, the selection of an appropriate decontamination method is an extremely important stage in the process of designing materials applied for packaging. The expected effect can also be achieved by the appropriate preparation of the material for sterilization, process conditions and subsequent storage if certain provisions are followed [2].

Chitosan is the polysaccharide which found widely used in biomaterials as it may be isolated from food industry byproducts [3]. It is safe and nontoxic, thereby, it may have contact with human tissues. The main disadvantage of chitosan is its low stability. It provides the need to cross-link chitosan by use cross-linkers [4]. Different compounds had been already proposed as effective cross-linkers for chitosan [5,6,7]. To not change the valuable biological properties of chitosan natural compounds as potential chitosan cross-linkers has been searched.

Polyphenols are natural compounds which are nowadays considered as safe for medical application [8]. They are able to form strong hydrogen bonds with polymers, thereby, are considered as effective cross-linkers for polysaccharides and proteins [9]. Various polyphenols had been already studied as chitosan crosslinkers as tannic acid [10], gallic acid [11], ellagic acid [12], caffeic acid [13], ferulic acid [14], etc. The novelty aspect concerns studying this influence of UVC irradiation on chitosan-based films modified by phenolic acids which may function as chitosan cross-linkers as well as antioxidant agents [15,16]. Chitosan/phenolic acid-based materials may find potential applications in food technology, as encapsulating agents, biomaterials, bioadsorbents or coatings [17,18,19]. In this study, we have made an attempt to determine the influence of UVC light on the properties of thin films obtained from chitosan modified by different phenolic acids: ferulic, caffeic and tannic acid.

## 2. Results

### 2.1. Fourier Transform Infrared Spectroscopy—Attenuated Total Reflectance (FTIR–ATR)

The spectra obtained for the chitosan-based films modified by phenolic acids are of a similar shape (Figure 1). All the characteristic peaks for chitosan are observed. A strong band in the 3329 cm^−1^ region corresponds to N-H and O-H stretching. The bands at around 2921 and 2851 cm^−1^ can be attributed to symmetric and asymmetric stretching of C-H, respectively. All these bands are characteristic of polysaccharides. A peak at 1641 cm^−1^ is observed on each spectrum, which confirms the presence of residual N-acetyl groups (C=O stretching of amide I) and 1321 cm^−1^ (C-N stretching of amide III). Furthermore, the peak corresponds to N-H bending of amide II to the one observed at 1531 cm^−1^. The CH_2_ bending and CH_3_ symmetrical deformations presence corresponds to the band at 1373 and 1314 cm^−1^. The band at 1062 cm^−1^ corresponds to the C-O stretching [20]. All the peaks are present in the spectra of the pure chitosan films as well as those modified by phenolic acids with no difference. It suggests that only hydrogen bonds are formed between chitosan and phenolic acid and they do not cause structural changes. When compared, the spectra of each film before and after irradiation present no significant changes resulting from exposure to UVC light. It allows making an assumption that the proposed sterilization method by UVC is safe and does not bring about chitosan/phenolic acid structural changes.

### 2.2. Scanning Electron Microscopy (SEM)

Scanning electron microscope images of the films cross-sections at the magnification of 10,000× are shown in Figure 2. It was found that the addition of phenolic acids changes the films structure when compared to that of the pure chitosan sample (images A,D,G,J). A film obtained from chitosan is porous and its morphology is not homogeneous. It is typical of polysaccharide films obtained by the solvent evaporation technique [21]. The microstructure of the films obtained from chitosan modified by phenolic acid is characterized by greater homogeneity, with no presence of visible pores. We assume that the cross-linking effect of phenolic acids influences changing the chitosan structure by hydrogen bonds formation. In each sample, the surface is smooth and flat, without any cracks. After irradiation, we observed small crashes in the CTS+CA and CTS+TA films.

### 2.3. Atomic Force Microscopy (AFM)

As can be clearly seen, the surface topography changes result from the material composition modification as well as exposure to UV light. The surface properties of materials are important when considering their application since they affect the interactions between microorganisms and materials surface [22]. The addition of phenolic acids to chitosan causes a decrease in roughness parameters, both Ra and Rq (Table 1). It may be observed that the films exposure to UVC for 1 h increases the surface roughness in each kind of sample. However, 2 h long irradiation brought about the opposite effect, i.e., roughness decreases in comparison to that of the non-irradiated samples. When TA and CA are added, the films surface presents the greatest smoothness (Figure 3).

### 2.4. Differential Scanning Calorimeter (DSC)

For each kind of sample, the first peak is observed at the temperature value in the range 81–93 °C, in dependence on a material composition (Table 2). The phenolic acids addition reduces the T_1_ values; the lowest temperature was observed for chitosan with tannic acid. The enthalpy for this process is in the range 0.84–1.06 mW/mg. The positive ΔH values suggest that the processes are endogenic; thereby, they are a consequence of the material decomposition. There are no significant temperature and enthalpy changes in samples irradiated for 1 and 2 h. Interesting observations can be made when the temperature and enthalpy for the second peak are compared. For pure chitosan, the second peak is observed at the temperature around 191 °C with the enthalpy 0.3684 mW/mg. In the case of films containing phenolic acids mentioned above, the peak is not observed for nonirradiated materials and after 1 h exposure to UVC. However, it should be emphasized that for films after 2 h of UVC irradiation, the second peak occurs in the temperature range 124 °C for chitosan with caffeic acid, 174 °C with ferulic acid and 196 °C with tannic acid, each with ΔH > 0.

### 2.5. Mechanical Properties

Mechanical properties are important when considering thin films use as packaging materials. Mechanical parameters as Young Modulus, maximum tensile strength and elongation at break were determined (Figure 4) with the use of the universal testing machine. First of all, it may be noticed that the phenolic acids addition affected the Young Modulus. The highest Emod value was observed for the material composed of chitosan and ferulic acid. Secondly, the irradiation by UVC slightly modified the Young Modulus of CTS and CTS+FA (after 1 h) as well as CTS+FA and CTS+TA (after 2 h). For chitosan, it increased twice after 1 h as a result of the photocross-linking process. However, in the case of films modified by phenolic acids, a decrease in Young Modulus is clearly seen. However, different correlations may be noticed for chitosan. Where the irradiation of pure chitosan for 1 and 2 h results in the improvement of σmax, the elongation at break of films obtained from CTS+TA is higher, but for CTS+FA and CTS+CA dl is lower; however, the UVC radiation does not cause any significant changes.

### 2.6. Surface Free Energy

The contact angle for glycerin as the hydrophilic solution was measured to determine the wettability of the film surface. The UVC irradiation decreases the wettability of the films based on chitosan with and without phenolic acids. The dangling bonds are exposed on the material surface and determine the surface free energy (Table 3) which controls the cells-material interactions. To better observe adhesion to the surface, is should be minimalized. High surface free energy inhibits the cell-material interactions. A polar component gives information about wetting of the solid by a liquid [23]. The addition of phenolic acids to chitosan results in the increase in the polar component which suggests that the hydrophilicity of film increases. The presence of many hydroxyl groups in the phenolic acids structure indicates the hydrophilicity change. In general, there is no constant trend to change surface parameters in dependence on the type of phenolic acid. The surface free energy slightly increases when a sample is exposed to UVC light. Thereby, we may assume that 1 and 2 h irradiation do not cause changes in the surface properties including surface free energy, dispersive and polar component.

### 2.7. Bacterial Growth Inhibition

The addition of phenolic acids into chitosan films does not significantly improve the *S. aureus* growth inhibition, and bacteria multiply at a similar rate (Table 4). The application of the irradiation process improves the antibacterial properties of the following films: CTS+TA and CTS+FA both after 1 h and 2 h as a significant slowdown in the bacterial multiplication rate is observed after 3 h.

In the case of *E. coli*, chitosan-based films containing caffeic and ferulic acid are characterized by a significantly greater bacterial growth inhibition in comparison to that of pure chitosan films (Table 5). The irradiation process also improves the antibacterial properties of CTS+CA and CTS+FA after 1 h irradiation and CTS+CA, CTS+TA and CTS+FA after 2 h exposition to UVC against *S. aureus*. The CTS+FA film shows particularly favorable antibacterial properties as after 2 h of the experiment the bacteria multiplication slows down almost twice (2.67 vs. 1.39 iMS).

When analyzing the obtained bacterial growth inhibition results, we can assume that irradiation positively influences the antibacterial properties of the obtained polymeric films. The CTS+FA shows the greatest antibacterial properties against the two tested bacteria. Then, CTS+TA is most effective in the case of *S. aureus* or CTS+CA—in the case of *E. coli*.

### 2.8. Adhesion of Bacteria to the Film Surface

The surface of films after 2 h irradiation was smooth and homogeneous with no defects. To evaluate whether the sterilization process somehow affects the biofilm formation on the films surface, SEM observations were made (Figure 5). The *S. aureus* biofilm was found on the surface of each type of film, however, for CTS+TA, it was much less developed. The *E. coli* biofilm was not observed on the CTS+TA surface and also for CTS+FA its formation was weakened. Thereby, we assume that CTS+TA is characterized by the most favorable properties for inhibition of the bacterial adhesion to the surface.

By comparing the two bacterial experiments, we can assume that CTS+FA after 2 h irradiation shows the most effective properties for inhibiting bacteria in a liquid solution (up to 4 h), but CTS+TA shows the most favorable bacteria inhibition to the surface (up to 14 days).

## 3. Discussion

Phenolic acids act as chitosan cross-linkers, which has already been reported in the literature [24,25,26]. Moreover, they have interesting active properties, antibacterial and antiviral for instance [27]. After addition into chitosan, phenolic acids interact with amine and hydroxyl groups of the polymer chain by hydrogen bonds [28] which causes significant changes in the material structure as well as its properties. A short time UVC light application accompanied by the material itself sterilization contributes to a more homogenous structure formation in the films as well as pores elimination. It may be related to a photocrosslinking process occurring during chitosan sample irradiation by UVC [29]. On the other hand, we have found essential changes in the CTS+TA and CTS+CA films surface after 2 h irradiation, such as appearance of new pores. We assume it may be caused by the photodegradation process and such a phenomenon has previously been observed [30]. Our results may be associated with the phenolic acids chemical structures in which especially tannic acid bears much more OH groups sensitive to photodegradation [31]. After exposure to UVC, the films surface remains flat and smooth, and even the roughness parameters are reduced. Similar results were obtained by Kowalonek [32,33] and Chełminiak-Dudkiewicz [34].

As a result of phenolic acids addition to chitosan, an improvement in mechanical parameters is observed. During irradiation with UVC light, two processes are competitive: photocrosslinking and photodegradation. It may be noticed that for films obtained from pure chitosan mainly the photocrosslinking process occur. After phenolic acids addition, only the photodegradation process was observed as the exposure of chitosan functional groups is lower than in the case of chitosan without additives. Nevertheless, only the ferulic acid addition results in a significant decrease in mechanical parameters, because those films especially degrade upon exposure to UVC.

Generally, the addition of phenolic acids does not cause significant changes in films surface properties. The results show that groups which reveal affinity to water are surface-oriented and change the surface properties. The wettability of each type of film decreased after the UVC irradiation. The dispersive component value is much higher than of a polar component despite the polar groups’ presence. Irradiation is not influenced by the surface free energy as well as the value of polar and dispersive components.

Different thermal behaviors of pure chitosan and chitosan/phenolic acid were previously successfully evaluated by the DSC technique [35]. A thermal analysis showed that all the changes which resulted from the samples heating are endothermal (ΔH > 0). DSC thermograms showed differences in films with phenolic acids after 2 h irradiation. In the case of films of chitosan with each phenolic acid, an additional peak is observed. Thereby, we may assume that the same changes occur after exposure to UVC and may be associated with the degradation processes. The UVC light may change the intermolecular hydrogen bonds orientation [36].

We observed better antimicrobial properties of chitosan films modified by phenolic acids against both, Gram-positive and Gram-negative bacteria. Phenolic acids have antimicrobial properties what have been already reported [37]. Lee at al. [38] showed that chitosan/gallic acid show antimicrobial activity against food pathogens. Božič et al. [39] confirmed antimicrobial properties of chitosan/caffeic acid materials. The antibacterial activity against a wide range of foodborne pathogens and spoilage bacteria was proven for chitosan/ferulic acid films by Chatterjee et al. [40]. Our research confirmed that the addition of phenolic acids improves the antibacterial activity of chitosan-based films against *S. aureus* and *E. coli*. Our results are in line with the study carried out by Wang et al. [41]. They determined the CTS+CA as the most effective bio-based food packaging. Moreover, the exposure of the films based on chitosan/ellagic acid to intense UV radiation did not alter any of their properties [16,42]. Our previous studies of UVC influence on the chitosan/tannic acid films showed that it modifies the material properties. However, we did not consider their antibacterial activity [43].

In the present experiment, the influence of UVC on the antibacterial properties of films obtained from chitosan/phenolic acids has been studied. We have found that this process may contribute to the improvement of antibacterial properties of films. In our opinion, the exposure to UVC light influenced the hydrogen bonds of the obtained materials and caused microstructure changes, especially after prolonged exposure. The most effective material to offer as food packaging is, in our opinion, chitosan with ferulic acid, as it exhibits the most effective antimicrobial activity. Moreover, it should be emphasized that the biofilm formation on its surface was not observed.

Summarizing, the UVC radiation seems to be an effective and safe method for the sterilization of chitosan/phenolic acids thin films, but it may also contribute to their smoothness improvement, a more porous microstructure formation as well as the antimicrobial properties activation in those materials. All these improvements are exceptionally beneficial for potential applications of films as food packaging. However, longer irradiation (2 h) causes more degenerative changes in film structures, which adversely affects their mechanical properties.

## 4. Materials and Methods

### 4.1. Materials

Chitosan (CTS, deacetylation degree: 78%, 1.8 × 106 D) and phenolic acids (tannic acid—TA, Mv = 1701.2 g/mol; ferulic acid—FA, trans-ferulic acid, >99%, Mv = 194.19 g/mol; and caffeic acid—CA, >98%, Mv = 180.16 g/mol) were purchased from Sigma-Aldrich (Poznan, Poland). Acetic acid was purchased from POCH (Gliwice, Poland).

### 4.2. Samples Preparation

Chitosan was dissolved in 0.1 M acetic acid at 2% concentration. Phenolic acids were also dissolve in 0.1 M acetic acid, at 1% concentration, each compound separately. A chitosan solution was mixed with a magnetic stirrer with 10 *v*/*v* phenolic acid solutions addition. Mixtures (40 mL) were then placed in plastic holders (10 cm × 10 cm) to evaporate the solvent (room conditions, 72 h).

Thin films were exposed to UVC light at 254 nm wavelength (ULTRAVIOL NBV 15 lamp, intensity: 18 W/m^2^) for 1 and 2 h. Films were irradiated in the distance of 5 cm from the lamp. Samples without the UV exposure were left as control.

### 4.3. Fourier Transform Infrared Spectroscopy—Attenuated Total Reflectance (FTIR–ATR)

FTIR-ATR spectra were performed for each type of sample in the range 4000–650 cm^−1^ with the Nicolet iS10 spectrometer (Thermo Fisher Scientific Inc., Waltham, MA, USA) equipped with a Ge single crystal. The spectra were recorded at the resolution of 4 cm^−1^ and 64 scans in the wavenumber range 600–4000 cm^−1^, and normalized. They were found in the absorbance mode.

### 4.4. Scanning Electron Microscopy (SEM)

A Scanning Electron Microscope (SEM; LEO Electron Microscopy Ltd., England) was used to observe the cross-section morphology of the obtained films. SEM was also used to observe the bacteria adhered to the material surface. In the both analyses, films were sputter-coated with gold, prior to the observation.

### 4.5. Atomic Force Microscopy (AFM)

Surface roughness was analyzed at room temperature with the use of a microscope with a scanning SPM probe of the NanoScope MultiMode type (Veeco Metrology, Inc., Santa Bar-bara, CA, USA) which operated in a tapping mode. Films (1 cm × 1 cm) were prepared and underwent the analysis. Surface roughness was determined by measuring two parameters (n = 5)—the root-mean-square (Rq) roughness and the arithmetic mean (Ra) within the Nanoscope v6.11 software (Bruker Optoc GmbH, Ettlingen, Germany).

### 4.6. Differential Scanning Calorimeter (DSC)

Differential scanning calorimetry measurements were carried out with differential scanning calorimeter equipment (NETZSCH Phoenix DSC 204 F1) at the heating rate of 10 °C/min, temperature range from 20 to 250 °C in nitrogen atmosphere with the flow of 40 mL/min. The samples (n = 5, weight 1.0–1.5 mg) were placed in the aluminum measuring pans.

### 4.7. Mechanical Properties

The mechanical properties were measured using a Universal Testing Machine (Z.05, Zwick/Roell, Ulm, Germany). The measurements (n = 10) ware carried out with the parameters of the initial force at 0.1 MPa and crosshead speed fixed at 5 mm/min. The Young Modulus, maximum tensile strength and elongation at break were calculated with the testXpert II program.

### 4.8. Surface Free Energy

Surface free energy—IFT(s), its polar—IFT(s,P) and dispersive—IFT(s,D) components can be calculated by the contact angle measurement. In this measurement, the non-covalent forces between the liquid and film surface are formed by Owens-Wendt method [44]. The contact angles of the liquids (glycerin and diiodomethane) were measured at a constant temperature value, using a goniometer equipped with a drop shape analysis system (DSA 10 Control Unit, Krüss, Germany).

### 4.9. Antimicrobial Activity

Bacterial growth inhibition was checked by measuring the cultured bacterial broth turbidity according to McFarland standards [45] with an assumption that there is a direct relation between the solution turbidity and the number of bacteria, and 1 McFarland index (iMS) corresponds to 3 × 10^8^ CFU/mL. Two bacterial strains were used for the tests: *Staphylococcus aureus* (ATCC 25923) and *Escherichia coli* (ATCC 35218), selected as various Gram groups representatives [46]. The study covered the tested films incubation (n = 3) in 2 mL of the bacterial solution and its optical density measurement with DensiChEK Plus (BioMerieux, Montreal, QC, Canada). The maximum measuring range of the device is 4 iMS; hence, the readings lasted 4 h. The bacteria were suspended in a Trypticase Soy Broth (Merck, Darmstadt, Germany), incubated at 37 °C, and their initial concentration for the tests was 0.3 iMS. Furthermore, the bacteria adhesion degree to the films surface was evaluated. The tests were performed by the specimens immersion in 3 mL of the above mentioned bacterial solution with 1 × 10^8^ CFU/mL inoculum (n = 3) and 14 days long incubation at 37 °C. A control sample was incubated in a solution without the addition of bacteria.

### 4.10. Statistical Analysis

Statistical analysis of the data was performed using commercial software (SigmaPlot 14.0, Systat Software, San Jose, CA, USA). The Shapiro–Wilk test was used to assess the normal distribution of the data. All of the results were statistically analyzed using one-way analysis of variance (one-way ANOVA) and are presented as a mean ± standard deviation (SD). Multiple comparisons between means were performed using the Student t-test with the statistical significance set at *p* < 0.05.

## 5. Conclusions

The addition of phenolic acids improves the physicochemical properties of chitosan-based films as they act as cross-linkers. Between them and chitosan, hydrogen bonds are formed. Moreover, films with phenolic acids showed better antimicrobial activity against both, Gram-positive and Gram-negative bacteria. Furthermore, the inhibition of biofilm formation was observed. Based on the obtained results, we confirmed that the sterilization of chitosan/phenolic acids films by the exposure to UVC light is effective. Both, the physicochemical properties of materials before and after exposure as well as their antimicrobial activity were compared. Chitosan composed with ferulic acid showed the most suitable properties required for food-packaging. Comparing the material features, we observed that 2 h exposure may initiate the photodegradation process. Hence, we recommend 1 h exposure as a standard sterilization process of food-packaging materials composed of chitosan with phenolic acids addition.

## Figures and Tables

**Figure 1 ijms-22-06472-f001:**
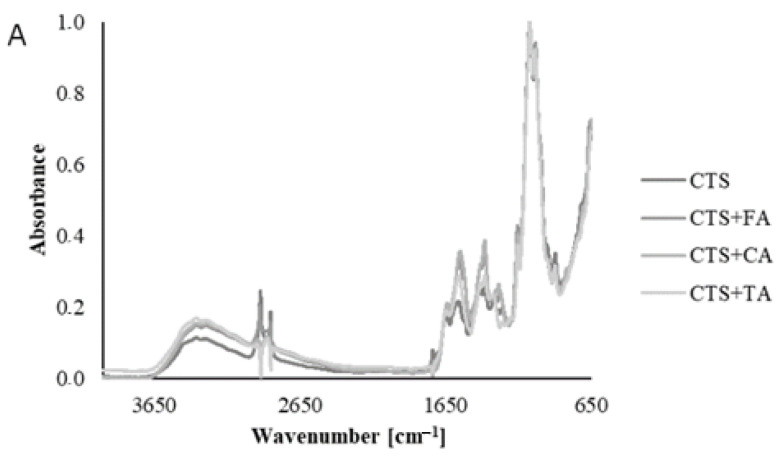
FTIR-ATR spectra of (**A**) CTS with and without FA, CA, TA and before and after irradiation of CTS (**B**), CTS+FA (**C**), CTS+CA (**D**), CTS+TA (**E**) for 1 and 2 h.

**Figure 2 ijms-22-06472-f002:**
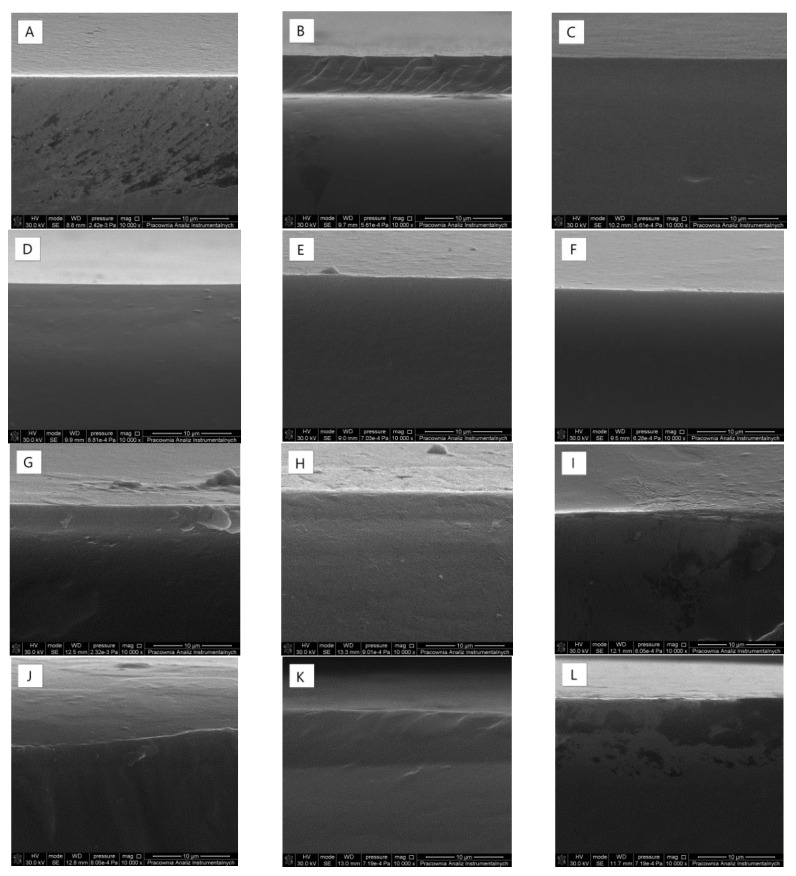
SEM images of cross-section morphology of the films (chitosan (**A–C**) modified by ferulic acid (**D–F**), caffeic acid (**G–I**) and tannic acid (**J**–**L**)) before and after 1 and 2 h irradiation (the presented images are representative for 5 specimens).

**Figure 3 ijms-22-06472-f003:**
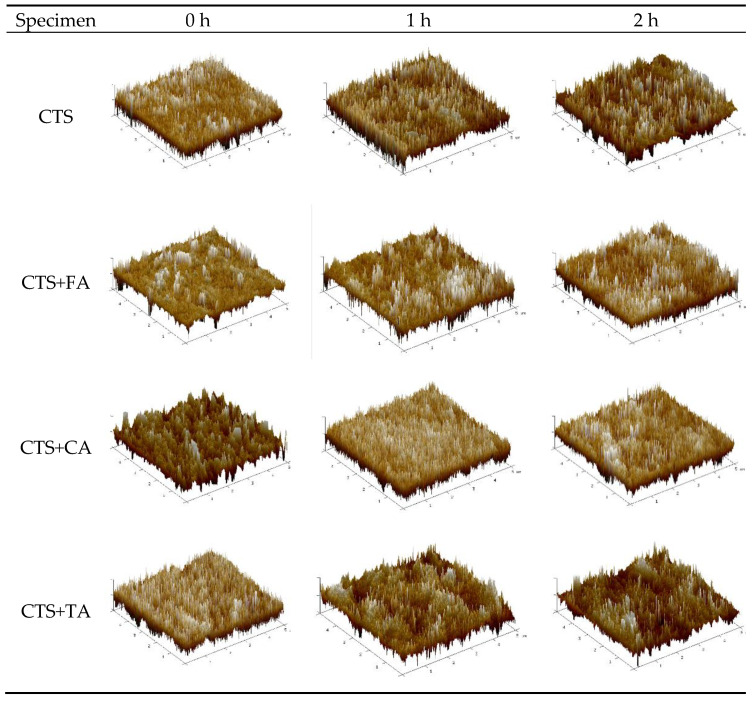
AFM three-dimensional images (5 × 5 µm) of chitosan films modified by ferulic, caffeic and tannic acid before and after 1 and 2 h irradiation (the presented images are representative for 5 specimens).

**Figure 4 ijms-22-06472-f004:**
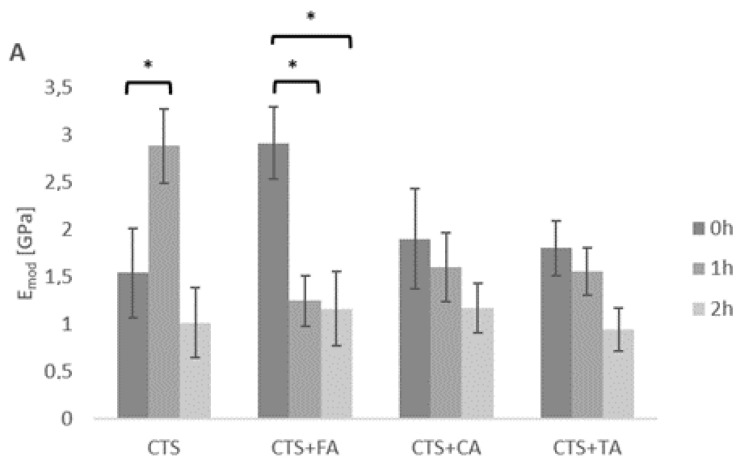
The Young Modulus (**A**), maximum tensile strength (**B**) and elongation at break (**C**) of films based on chitosan (CTS) with ferulic (FA), caffeic (CA) and tannic acid (TA) of samples non-irradiated and irradiated for 1 and 2 h (n = 10, mean ± SD, * significantly different between the groups—*p* < 0.05).

**Figure 5 ijms-22-06472-f005:**
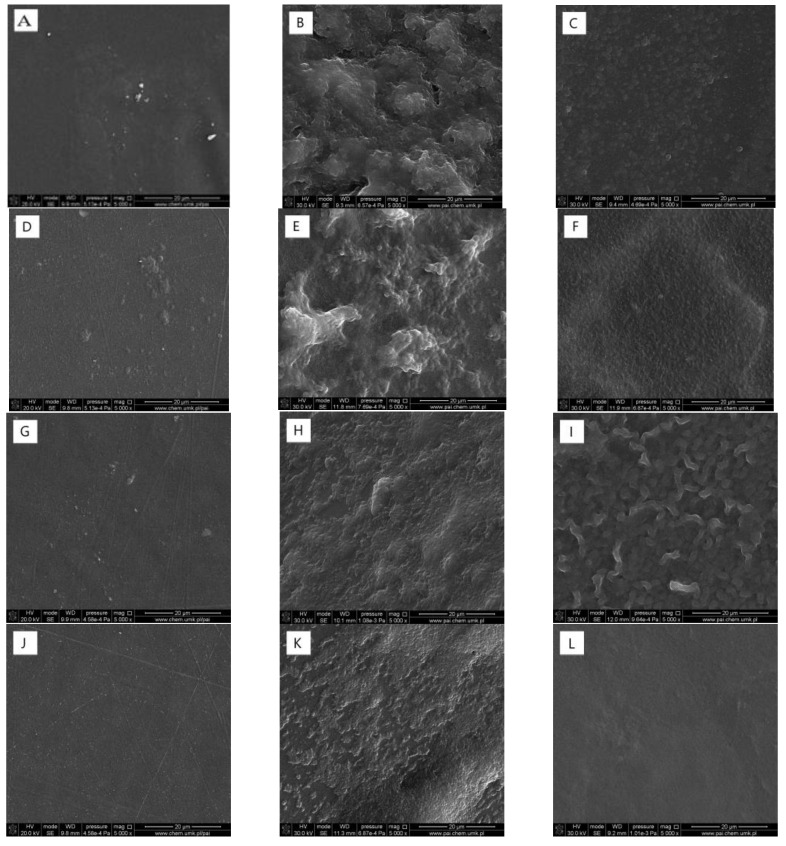
Comparison of bacterial adhesion to the films surface after 14 days of incubation in a bacterial suspension for Figure 2. h: CTS (**A**–**C**), CTS+FA (**D**–**F**), CTS+CA (**G**–**I**), CTS+TA (**J**–**L**) as control (**A**,**D**,**G**,**J**), *Staphylococcus aureus* (**B**,**E**,**H**,**K**), *Escherichia coli* (**C**,**F**,**I**,**L**) (SEM 5000×; the presented pictures are representative for 3 experiments).

**Table 1 ijms-22-06472-t001:** Roughness parameters (Ra and Rq) of chitosan films modified by ferulic, caffeic and tannic acid before and after 1 and 2 h irradiation (n = 5; * significantly different from non-irradiated—*p* < 0.05; ^#^ significantly different from control—CTS—*p* < 0.05).

Specimen	Ra [nm]			Rq [nm]		
	0 h	1 h	2 h	0 h	1 h	2 h
CTS	74.1 ± 0.2	99.4 ± 0.1 *	65.9 ± 0.3 *	88.3 ± 0.2	122.0 ± 0.1 *	79.5 ± 0.2 *
CTS+FA	59.9 ± 0.1 ^#^	85.1 ± 0.2 *^,#^	48.5 ± 0.2 *^,#^	72.2 ± 0.3 ^#^	103.0 ± 0.2 *^,#^	61.1 ± 0.3 *^,#^
CTS+CA	23.7 ± 0.1 ^#^	30.4 ± 0.2 *^,#^	16.5 ± 0.2 *^,#^	30.8 ± 0.2 ^#^	37.4 ± 0.3 *^,#^	20.8 ± 0.4 *^,#^
CTS+TA	22.5 ± 0.2 ^#^	29.9 ± 0.3 *^,#^	19.5 ± 0.4 *^,#^	28.6 ± 0.2 ^#^	39.0 ± 0.3 *^,#^	23.3 ± 0.3 *^,#^

**Table 2 ijms-22-06472-t002:** The maximum temperature of the thermal process (T) and enthalpy of the processes (ΔH) measured during the samples heating by differential scanning calorimetry (n = 5).

Specimen	T_1_ [°C]			ΔH [mW/mg]			T_2_ [°C]			ΔH [mW/mg]		
	0 h	1 h	2 h	0 h	1 h	2 h	0 h	1 h	2 h	0 h	1 h	2 h
CTS	93.3	77.6	80.3	0.974	1.274	0.871	191.8	193.8	-	0.368	0.328	-
CTS+FA	86.9	90.4	77.1	1.062	0.980	1.024	-	-	174.0	-	-	0.221
CTS+CA	86.2	80.2	80.9	0.841	1.036	0.666	-	-	124.3	-	-	0.733
CTS+TA	81.1	80.6	83.2	0.888	1.068	0.792	-	-	196.4	-	-	0.203

**Table 3 ijms-22-06472-t003:** The contact angle for glycerin (θ^G^), for diiodomethane (θ^I^), the surface free energy (IFT(s)), its polar (IFT(s,D)) and dispersive (IFT(s,D)) components of films based on chitosan (CTS) with ferulic (FA), caffeic (CA) and tannic acid (TA) of samples non-irradiated and irradiated for 1 and 2 h (n = 5; * significantly different from non-irradiated—*p* < 0.05; # significantly different from control—CTS—*p* < 0.05).

Specimen	θ^G^ [°]	θ^I^ [°]	IFT(s) [mJ/m^2^]	IFT(s,D) [mJ/m^2^]	IFT(s,P) [mJ/m^2^]
non-irradiated					
CTS	89.80 ± 3.98	59.54 ± 1.18	28.39 ± 0.44	27.02 ± 0.30	1.36 ± 0.15
CTS+FA	84.37 ± 3.07 ^#^	61.50 ± 0.49 ^#^	27.68 ± 0.32 ^#^	24.40 ± 0.15 ^#^	3.29 ± 0.17 ^#^
CTS+CA	94.15 ± 1.55	62.68 ± 0.49 ^#^	26.71 ± 0.16 ^#^	25.95 ± 0.12 ^#^	3.76 ± 0.04 ^#^
CTS+TA	81.60 ± 1.09 ^#^	56.33 ± 0.75 ^#^	30.58 ± 0.24 ^#^	27.15 ± 0.17	3.44 ± 0.07 ^#^
1 h					
CTS	80.14 ± 1.27 *	54.28 ± 2.13 *	31.78 ± 0.62 *	28.13 ± 0.48 *	3.64 ± 0.14 *
CTS+FA	77.60 ± 1.92 *	52.34 ± 0.72 *	33.08 ± 0.31 *^,#^	28.76 ± 0.18 *^,#^	4.32 ± 0.13 *^,#^
CTS+CA	88.58 ± 3.54 *^,#^	63.48 ± 0.69 ^#^	26.26 ± 0.35 *^,#^	24.07 ± 0.19 *^,#^	2.19 ± 0.16 *^,#^
CTS+TA	78.37 ± 1.03 *	54.77 ± 1.20	31.82 ± 0.37 *	27.38 ± 0.27 ^#^	4.44 ± 0.10 *^,#^
2 h					
CTS	71.47 ± 0.31 *	53.08 ± 0.53 *	34.16 ± 0.17 *	26.80 ± 0.12	7.36 ± 0.05 *
CTS+FA	69.67 ± 0.90 *^,#^	49.46 ± 1.81 *^,#^	36.10 ± 0.57 *^,#^	28.64 ± 0.41 *^,#^	7.46 ± 0.17 *^,#^
CTS+CA	88.00 ± 0.95 *^,#^	60.42 ± 0.34 *^,#^	27.92 ± 0.12 *^,#^	25.98 ± 0.08 ^#^	1.94 ± 0.04 *^,#^
CTS+TA	78.05 ± 1.46 *^,#^	54.84 ± 0.58 *	31.39 ± 0.23 *^,#^	27.99 ± 0.14 *^,#^	3.39 ± 0.09 *^,#^

**Table 4 ijms-22-06472-t004:** The *Staphylococcus aureus* growth inhibition during incubation for a specific period of time with the tested films (n = 3; max. SD = 0.03; * significantly different from non-irradiated—*p* < 0.05; # significantly different from control—CTS—*p* < 0.05).

McFarland Standard Values Specifying the Number of *Staphylococcus aureus* Bacteria								
Specimen	Non-Irradiated							
	CTS		CTS+CA		CTS+TA		CTS+FA	
Time [h]	iMS	App. number of bacteria	iMS	App. number of bacteria	iMS	App. number of bacteria	iMS	App. number of bacteria
0	0.30	0.9 × 10^8^	0.30	0.9 × 10^8^	0.30	0.9 × 10^8^	0.30	0.9 × 10^8^
0.5	0.50	1.5 × 10^8^	0.50	1.5 × 10^8^	0.48	1.4 × 10^8^	0.48	1.4 × 10^8^
1	0.81	2.4 × 10^8^	0.81	2.4 × 10^8^	0.80	2.4 × 10^8^	0.82	2.5 × 10^8^
2	1.89	5.7 × 10^8^	1.90	5.7 × 10^8^	1.85	5.6 × 10^8^	1.82 ^#^	5.5 × 10^8^
3	2.51	7.5 × 10^8^	2.52	7.6 × 10^8^	2.46	7.4 × 10^8^	2.47	7.4 × 10^8^
4	>4	>12 × 10^8^	>4	>12 × 10^8^	>4	>12 × 10^8^	>4	>12 × 10^8^
Specimen	irradiated 1 h							
	CTS		CTS+CA		CTS+TA		CTS+FA	
Time [h]	iMS	App. number of bacteria	iMS	App. number of bacteria	iMS	App. number of bacteria	iMS	App. number of bacteria
0	0.30	0.9 × 10^8^	0.30	0.9 × 10^8^	0.30	0.9 × 10^8^	0.30	0.9 × 10^8^
0.5	0.50	1.5 × 10^8^	0.50	1.5 × 10^8^	0.47	1.4 × 10^8^	0.51	1.5 × 10^8^
1	0.79	2.4 × 10^8^	0.80	2.4 × 10^8^	0.82	2.5 × 10^8^	0.82	2.5 × 10^8^
2	1.91	5.7 × 10^8^	1.87	5.6 × 10^8^	1.83	5.5 × 10^8^	1.82 ^#^	5.5 × 10^8^
3	2.49	7.5 × 10^8^	2.49	7.5 × 10^8^	2.37 *^,#^	7.1 × 10^8^	2.28 *^,#^	6.8 × 10^8^
4	>4	>12 × 10^8^	>4	>12 × 10^8^	>4	>12 × 10^8^	>4	>12 × 10^8^
Specimen	irradiated 2 h							
	CTS		CTS+CA		CTS+TA		CTS+FA	
Time [h]	iMS	App. number of bacteria	iMS	App. number of bacteria	iMS	App. number of bacteria	iMS	App. number of bacteria
0	0.30	0.9 × 10^8^	0.30	0.9 × 10^8^	0.30	0.9 × 10^8^	0.30	0.9 × 10^8^
0.5	0.51	1.5 × 10^8^	0.47	1.4 × 10^8^	0.48	1.4 × 10^8^	0.51	1.5 × 10^8^
1	0.82	2.5 × 10^8^	0.81	2.5 × 10^8^	0.81	2.4 × 10^8^	0.81	2.4 × 10^8^
2	1.86	5.6 × 10^8^	1.91	5.7 × 10^8^	1.83	5.5 × 10^8^	1.81 ^#^	5.4 × 10^8^
3	2.52	7.6 × 10^8^	2.51	7.5 × 10^8^	2.34 *^,#^	7.0 × 10^8^	2.20 *^,#^	6.6 × 10^8^
4	>4	>12 × 10^8^	>4	>12 × 10^8^	>4	>12 × 10^8^	>4	>12 × 10^8^

**Table 5 ijms-22-06472-t005:** The *Escherichia coli* growth inhibition during incubation with the tested films (n = 3; max. SD = 0.03; * significantly different from non-irradiated—*p* < 0.05; # significantly different from control—CTS—*p* < 0.05).

McFarland Standard Values Specifying the Number of *Escherichia coli* Bacteria								
Specimen	Non-Irradiated							
	CTS		CTS+CA		CTS+TA		CTS+FA	
Time [h]	iMS	App. number of bacteria	iMS	App. number of bacteria	iMS	App. number of bacteria	iMS	App. number of bacteria
0	0.30	0.9 × 10^8^	0.30	0.9 × 10^8^	0.30	0.9 × 10^8^	0.30	0.9 × 10^8^
0.5	0.53	1.6 × 10^8^	0.51	1.5 × 10^8^	0.50	1.5 × 10^8^	0.52	1.6 × 10^8^
1	0.92	2.8 × 10^8^	0.97	2.9 × 10^8^	0.81 ^#^	2.4 × 10^8^	0.75 ^#^	2.3 × 10^8^
2	2.67	8.0 × 10^8^	2.45 ^#^	7.4 × 10^8^	2.62	7.9 × 10^8^	1.62 ^#^	4.9 × 10^8^
3	>4	>12 × 10^8^	>4	>12 × 10^8^	>4	>12 × 10^8^	2.97	8.9 × 10^8^
4							>4	>12 × 10^8^
Specimen	irradiated 1 h							
	CTS		CTS+CA		CTS+TA		CTS+FA	
Time [h]	iMS	App. number of bacteria	iMS	App. number of bacteria	iMS	App. number of bacteria	iMS	App. number of bacteria
0	0.30	0.9 × 10^8^	0.30	0.9 × 10^8^	0.30	0.9 × 10^8^	0.30	0.9 × 10^8^
0.5	0.51	1.5 × 10^8^	0.52	1.6 × 10^8^	0.50	1.6 × 10^8^	0.51	1.5 × 10^8^
1	0.92	2.8 × 10^8^	0.89 *	2.7 × 10^8^	0.82 ^#^	2.5 × 10^8^	0.71 ^#^	2.1 × 10^8^
2	2.65	8.0 × 10^8^	2.33 *^,#^	7.0 × 10^8^	2.63	7.9 × 10^8^	1.46 *^,#^	4.4 × 10^8^
3	>4	>12 × 10^8^	>4	>12 × 10^8^	>4	>12 × 10^8^	2.85 *^,#^	8.6 × 10^8^
4							>4	>12 × 10^8^
Specimen	irradiated 2 h							
	CTS		CTS+CA		CTS+TA		CTS+FA	
Time [h]	iMS	App. number of bacteria	iMS	App. number of bacteria	iMS	App. number of bacteria	iMS	App. number of bacteria
0	0.30	0.9 × 10^8^	0.30	0.9 × 10^8^	0.30	0.9 × 10^8^	0.30	0.9 × 10^8^
0.5	0.51	1.5 × 10^8^	0.50	1.5 × 10^8^	0.50	1.5 × 10^8^	0.51	1.5 × 10^8^
1	0.94	2.8 × 10^8^	0.86 *	2.6 × 10^8^	0.82 ^#^	2.5 × 10^8^	0.69 *^,#^	2.1 × 10^8^
2	2.66	8.0 × 10^8^	2.3 *^,#^	6.9 × 10^8^	2.46 *^,#^	7.4 × 10^8^	1.39 *^,#^	4.2 × 10^8^
3	>4	>12 × 10^8^	>4	>12 × 10^8^	>4	>12 × 10^8^	2.32 *^,#^	7.0 × 10^8^
4							>4	>12 × 10^8^

## Data Availability

The data presented in this study are available on request from the corresponding author. The data are not publicly available due to project realization.

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
