# Peer review of "The Physicochemical and Antibacterial Properties of Chitosan-Based Materials Modified with Phenolic Acids Irradiated by UVC Light"

_ijms, 2021, doi:10.3390/ijms22126472_

Round 1

Reviewer 1 Report

The manuscript titled "The physicochemical and antibacterial properties of chitosan-based materials modified with phenolic acids irradiated by UVC light" written by Kaczmarek-Szczepańska et al. is overall well-written. The research topic is novel and of contemporary interest and application significance. The results provide directions for novel applications of chitosan and phenolic acids. It could be published in IJMS with some modifications as listed below to improve the clarity and readability of the manuscript.

Line 17 & 21: The abbreviations UVC, FTIR, SEM, AFM, DSC should be definied (spelled out) before using the abbreviations.

Line 42: "... the need to cross-ling chitosan..." The word "cross-ling" should be "cross-link"?

Line 60-63: (1) The figures are not clear enough. Please provide figures of higher resolution. (2) The figure panels could be aligned better.

Line 64: The abbreviations CTS, FA, CA, TA should be defined at the first appearance, instead of defining in the later Materials and Methods section.

Line 84, 198: Table 1 and Table 7 are basically figures, please present them in the format of Figures (e.g., Figure 2, Figure 5). In addition, please align the small figures (Now some are smaller, some are bigger).

Line 86, 101: "Oh" should be "0h" (Please check carefully - it seems you used the capital letter "O" instead of the number "0")

Line 129: The first "degree Celcius" symbol is not in the right format, please correct it. In addition, the format of this symbol in Results section (oC) is different from that in Materials and Methods section (°C), please check carefully and keep it consistent.

Figure 3A (y-axis), Table 5&6 (0,5 h), Line 284 (180,16 g/mol): Please keep the format of decimal separator consistent as the majority English-style decimal separator (e.g., 0.5 h; 180.16 g/mol). The French style (i.e., using a comma) is not wrong, but just that consistency is important.

Figure 3: Please align the three panels. 

Table 5, 6: (1) The first CTS, Time [h], MSi are bold while the others are not. Please keep it consistent. (2) The unit "MSi" in the tables does not correspond to the unit "iMS" for the McFarland index written in Section 4.9 (lime 336). Please use the correct format consistently.

Line 172, 176, 181, 184, 188, 195, 203, 205...: The scientific names (Genus species) must be italicized.

Line 216: "... the polymer chain by hydrogen [28]..." To the best of my knowledge, the word here should be "hydrogen bonds" or "hydrogen bonding"

Line 282: The abbreviation DD is not defined.

Author Response

Line 17 & 21: The abbreviations UVC, FTIR, SEM, AFM, DSC should be definied (spelled out) before using the abbreviations.

Thank you very much for the suggestion. It is now corrected.

Line 42: "... the need to cross-ling chitosan..." The word "cross-ling" should be "cross-link"?

Thank you very much. We absolutely agree. It is now corrected.

Line 60-63: (1) The figures are not clear enough. Please provide figures of higher resolution. (2) The figure panels could be aligned better.

Thank you very much for the suggestion. Figures are corrected.

Line 64: The abbreviations CTS, FA, CA, TA should be defined at the first appearance, instead of defining in the later Materials and Methods section.

All abbreviations are now defined at the first appearance. We defined mentioned abbreviations in the abstract.

Line 84, 198: Table 1 and Table 7 are basically figures, please present them in the format of Figures (e.g., Figure 2, Figure 5). In addition, please align the small figures (Now some are smaller, some are bigger).

Thank you very much for the suggestion. It is now corrected.

Line 86, 101: "Oh" should be "0h" (Please check carefully - it seems you used the capital letter "O" instead of the number "0")

Thank you very much. We corrected it.

Line 129: The first "degree Celcius" symbol is not in the right format, please correct it. In addition, the format of this symbol in Results section (oC) is different from that in Materials and Methods section (°C), please check carefully and keep it consistent.

Thank you very much for your comment. We corrected it.

Figure 3A (y-axis), Table 5&6 (0,5 h), Line 284 (180,16 g/mol): Please keep the format of decimal separator consistent as the majority English-style decimal separator (e.g., 0.5 h; 180.16 g/mol). The French style (i.e., using a comma) is not wrong, but just that consistency is important.

Thank you very much. We agree. It is now corrected.

Figure 3: Please align the three panels.

We appreciate your comment. It is now corrected.

Table 5, 6: (1) The first CTS, Time [h], MSi are bold while the others are not. Please keep it consistent. (2) The unit "MSi" in the tables does not correspond to the unit "iMS" for the McFarland index written in Section 4.9 (lime 336). Please use the correct format consistently.

Thank you for the comment. We corrected it.

Line 172, 176, 181, 184, 188, 195, 203, 205...: The scientific names (Genus species) must be italicized.

Thank you very much for the suggestion. It is now corrected.

Line 216: "... the polymer chain by hydrogen [28]..." To the best of my knowledge, the word here should be "hydrogen bonds" or "hydrogen bonding"

You are absolutely right. It is now corrected.

Line 282: The abbreviation DD is not defined.

We corrected it on “deacetylation degree”

Reviewer 2 Report

The Material and methods shoul be before Results and discussion.

FTIR-ATR - normalization data is missing, at which wavelength and then compare. If it was done that way then explai.

SFE - test liquids are not deffined. Water, diiodomethane or else.

In Table 4 list analogue contact angles measured by these liquids. Instrument calculate from measured contact angled. From the contact angle hydrophility can be determined as well. The question is lower contact angle or higher capilarity? Explian when the results are added.

Author Response

The Material and methods shoul be before Results and discussion.

Thank you very much for the comment. However, we used the Microsoft Word template from the journal website. According the regulations, “Materials and methods” section is after discussion part and before conclusion.

FTIR-ATR - normalization data is missing, at which wavelength and then compare. If it was done that way then explai.

Thank you very much for the comment. It is now corrected in the “materials and methods section”

SFE - test liquids are not deffined. Water, diiodomethane or else.

The measurement was carried out for glycerin and diiodomethane. It is  now written in the paper.

In Table 4 list analogue contact angles measured by these liquids. Instrument calculate from measured contact angled. From the contact angle hydrophility can be determined as well. The question is lower contact angle or higher capilarity? Explian when the results are added.

 Thank you very much for the suggestion. We added the results and discussion.